# 30-day mortality after hip fracture surgery: Influence of postoperative factors

**Juan F. Blanco**[1,2], **Carmen da Casa**[2]*, **Carmen Pablos-Hernández**[2,3], **Alfonso González-Ramírez**[2,3], **José Miguel Julián-Enríquez**[1,2], **Agustín Díaz-Álvarez**[2,4]

**1** Trauma and Orthopaedics Department, University Hospital of Salamanca, Salamanca, Spain, **2** Instituto de Investigación Biomédica de Salamanca (IBSAL), Salamanca, Spain, **3** Orthogeriatric Unit, University Hospital of Salamanca, Salamanca, Spain, **4** Anaesthesiology Department, University Hospital of Salamanca, Salamanca, Spain

* cdacasap@saludcastillayleon.es

## Abstract

### Purpose

The 30-day mortality rate after hip fracture surgery has been considered as an indirect indicator of the quality of care. The aim of this work is to analyse preoperative and postoperative factors potentially related to early 30-day mortality in patients over 65 undergoing hip fracture surgery.

### Methods

Prospective cohort study including all consecutive primary hip fracture patients over 65 admitted to Trauma and Orthopaedics department from January 1, 2018 to December 31, 2019. Bed-ridden, non- surgically treated patients, and high energy trauma or tumoral aetiology fractures were excluded. A total of 943 patients were eligible (attrition rate: 2.1%). Follow-up included 30-days after discharge. We noted the 30-day mortality after hip fracture surgery, analysing 130 potentially related variables including biodemographic, fracture-related, preoperative, and postoperative clinical factors. Qualitative variables were assessed by $\chi^2$, and quantitative variables by non-parametric tests. Odds ratio determined by binary logistic regression. We selected preventable candidate variables for multivariate risk assessment by logistic regression.

### Results

A total of 923 patients were enrolled (mean age 86.22±6.8, 72.9% women). The 30-day mortality rate was 6.0%. We noted significant increased mortality on men (OR = 2.381[1.371–4.136], p = 0.002), ageing patients ($OR_{year}$ = 1.073[1.025–1.122], p = 0.002), and longer time to surgery ($OR_{day}$ = 1.183[1.039–1146], p<0.001), on other 20 preoperative clinical variables, like lymphopenia (lymphocyte count <$10^3$/µl, OR = 1.842[1.063–3.191], p = 0.029), hypoalbuminemia (≤3.5g/dl, OR = 2.474[1.316–4.643], p = 0.005), and oral anticoagulant intake (OR = 2.499[1.415–4.415], p = 0.002), and on 25 postoperative clinical variables, like arrhythmia (OR = 13.937[6.263–31.017], p<0.001), respiratory insufficiency (OR = 7.002 [3.947–12.419], p<0.001), hyperkalaemia (OR = 10.378[3.909–27.555], p<0.001),

**Data Availability Statement:** There are both ethical and legal restrictions on sharing the original study datasets. The electronic health records data cannot be shared publicly because it consists of personal

information from which it is difficult to guarantee de-identification (Law 03/2018 from Spanish Government - BOE-A-2018-16673). There is a possibility of deductive disclosure of participants and therefore full data access through a public repository. The original datasets could only be made available under a new data sharing agreement with which includes: 1) commitment to using the data only for research purposes and not to identify any individual participant; 2) a commitment to securing the data using appropriate measures, and 3) a commitment to destroy or return the data after analyses are complete. For more information on data availability restrictions you can contact the ethics committee local IRB CEIm Area de Salud de Salamanca at comite.etico. husa@saludcastillayleon.es. Requests can be made to the corresponding author, who will connect the request to designated IRB representatives, and eventually send the information.

**Funding:** The authors received no specific funding for this work.

**Competing interests:** The authors have declared that no competing interests exist.

nutritional supply requirement (OR = 3.576[1.894–6.752], p = 0.021), or early arthroplasty dislocation (OR = 6.557[1.206–35.640], p = 0.029). We developed a predictive model for early mortality after hip fracture surgery based on postoperative factors with 96.0% sensitivity and 60.7% specificity (AUC = 0.863).

## Conclusion

We revealed that not only preoperative, but also postoperative factors have a great impact after hip fracture surgery. The influence of post-operative factors on 30-day mortality has a logical basis, albeit so far they have not been identified or quantified before. Our results provide an advantageous picture of the 30-day mortality after hip fracture surgery.

## Introduction

Treatment of hip fractures in the older population is a relevant part of the healthcare activity of trauma and orthopaedic services. In the last years, many factors have raised to improve the results of hip fracture treatment, like the early surgery rates or the establishment of healthcare multidisciplinary units, the so-called orthogeriatric units [1, 2]. Despite these improvements, mortality following a hip fracture is high [3]. The early mortality rate, defined as 30-day mortality rate after hospital discharge, has been considered as an indirect indicator of the quality of care [4]. It seems to be interesting to identify the more susceptible patients to early die and thereby increase the resources and efforts for their treatment. Many studies have looked into different factors that could be associated with early mortality after hip fracture [4–8]. Previous works show also differences in the results depending on the country where the study was carried out. These differences could, in part, also reflect differences in life-expectancy and healthcare systems [9]. The most frequently early mortality related factors include biodemographic factors, like age and gender, and clinical factors, like time to surgery or comorbidities [10]. Some recently developed models assessing early mortality after hip fracture showed limited application and no postoperative approach have been assessed [11, 12]. Although these factors could be less modifiable, knowing whether they can affect the early outcome would represent a great advantage for the older hip fracture patients' management.

The aim of this work is to analyse which factors—not only preoperative, but also postoperative and in-hospital-derived factors—could be related to 30-day mortality in patients older than 65-years-old who underwent surgery following a hip fracture at the University Hospital of Salamanca, and therefore establish a predictive model for early mortality so it could help to improve the in-hospital healthcare to this group of patients.

## Materials and methods

We design a prospective cohort study including all patients undergoing hip fracture surgery at the University Hospital of Salamanca from January 1st 2018 to December 31st 2019.

We included all patients over 65 with the main diagnosis of primary hip fracture. We excluded bed-ridden patients, non- surgically treated patients, and high energy trauma or tumoral aetiology fractures. In all cases, neuraxial anaesthesia was used [13]. A total of 943 patients were enrolled by signing informed consent, from whose clinical history we collected data for 130 variables. We performed on-site or telephonic 30-day follow up, noting down records for mortality. The attrition rate was 2.1%, leading a study population of 923 patients.

The study variables are grouped as biodemographic variables, like gender or place of residence; fracture-related variables, like the type of fracture or time to surgery; preoperative clinical variables, like comorbidities or admission laboratory data; and postoperative clinical variables, like surgical complications, medical complications, or walking ability at discharge.

### Ethics approval

The whole study was conducted following the Declaration of Helsinki and previously approved by the ethics committee for clinical research (CEIm) of the University Hospital of Salamanca (code: PI202001418). All participants (or their relatives) have given their written informed consent to participate.

### Statistics

Data collection was done using Microsoft® Office Access 2016 (Microsoft, Inc., Redmond, WA) and data analysis was done using IBM® SPSS Statics, version 25 (IBM, Inc., Armonk, NY).

Qualitative variables were analysed by contingency tables, and their statistical significance was assessed by $\chi^2$. Quantitative variables were analysed by mean, standard deviation, median, and range, and their statistical significance was assessed by non-parametric tests.

We performed a univariate risk assessment, estimating the odds ratio (OR) by binary logistic regression. For the multivariate analysis, we randomized patients at 50% to validate the results in half population. Seeking into previous analyses, we selected preventable candidate variables for multivariate risk assessment. We determined a predictive model for 30-days mortality, evaluated by logistic regression and ROC curve. We considered statistically significant $p \leq 0.05$ in all cases.

### Results

A total of 923 patients were enrolled and completed the 30-day follow-up. We noted a 30-day mortality rate of 6.0%, including 3.4% of in-hospital mortality rate.

Table 1 shows descriptive data for all biodemographic and fracture-related factors analysed in the study.

### Biodemographic variables

We noted a significantly higher rate of men in the early-mortality group (OR = 2.381 [1.371–4.136], p = 0.002), showing increasing proportion than the original population gender distribution. We also noted an increased risk for the 30-day mortality with the increasing age of patients (OR per year = 1.073 [1.025–1.122], p = 0.002), and a higher rate of patients over age 90, than patients under 90, in the 30-day mortality group (p = 0.006). We did not note significant differences regarding the place of residence or institution-living patients.

### Fracture-related variables

We did not note significant differences in the type of diagnosis or type of surgical procedure. We noted significant differences in the increasing time to surgery (OR per day = 1.183 [1.039–1146], p<0.001), noting down the protective effect for patients treated in the first 24h (OR = 0.329 [0.117–0.924], p = 0.035).

The complete analysed descriptive data for all preoperative and postoperative factors analysed in the study are on S1 Table.

**Table 1. Frequencies of biodemographic and fracture-related variables.**

| | Total population (n = 923) | Survival (n = 868) | 30-day mortality (n = 55) | p-value |
|---|---|---|---|---|
| **Biodemographic variables** | | | | |
| Female gender | 72.9% | 74.1% | 54.5% | **0.002** |
| Age (years) | 86.22 ± 6.80 | 86.05 ± 6.79 | 88.93 ± 6.48 | **0.006** |
| | 87 [65–103] | 87 [65–103] | 89 [73–100] | |
| Older than 85-years-old | 64.8% | 64.2% | 74.5% | 0.118 |
| Older than 90-years-old | 32.4% | 31.3% | 49.1% | **0.006** |
| Rural municipality (<12,500 inhabitants) | 48.2% | 48.6% | 41.8% | 0.328 |
| Institution-living (Nursing facility) | | | | |
| At admission | 32.8% | 33.0% | 29.1% | 0.550 |
| At discharge | 54.6% | 54.4% | 58.8% | 0.612 |
| Post-hospitalization | 32.1% | 31.8% | 39.1% | 0.458 |
| **Fracture-related variables** | | | | |
| Left side | 53.0% | 53.20% | 54.50% | 0.849 |
| Type of fracture | | | | 0.525 |
| Subcapital | 43.2% | 43.0% | 47.3% | |
| Basicervical | 4.7% | 4.7% | 3.6% | |
| Pertrochanteric | 45.8% | 45.7% | 47.3% | |
| Subtrochanteric | 6.3% | 6.6% | 1.7% | |
| Intracapsular fracture | 53.0% | 52.3% | 49.1% | 0.644 |
| Surgical intervention | | | | 0.482 |
| Osteosynthesis | 55.9% | 56.0% | 54.5% | |
| Partial Hip Replacement | 41.9% | 41.7% | 45.5% | |
| Total Hip Replacement | 2.2% | 2.3% | - | |
| Time to surgery (days) | 2.89 ± 2.57 | 2.80 ± 2.46 | 4.33 ±3.67 | **0.001** |
| | 3 [0–23] | 2 [0–23] | 4 [0–20] | |
| Delay <24 hours | 18.5% | 19.2% | 7.3% | **0.027** |
| Delay <48 hours | 36.1% | 36.9% | 23.6% | **0.048** |

Frequencies are shown in percentages and quantitative variables are described by mean ± standard deviation and median [range]. Significant differences on survival and early mortality comparison are marked in bold.

## Preoperative clinical variables

Among the lab preoperative clinical variables analysed, we noted significant differences on the incidence of lymphopenia (lymphocyte count $<10^3$/μl, OR = 1.842 [1.063–3.191], p = 0.029) and the albumin admission level (OR per g/dl = 0.507 [0.268–0.959], p = 0.037), establishing the critical point at albumin admission level ≤3.5g/dl (OR = 2.474 [1.316–4.643], p = 0.005).

We also noted significant differences in patients presenting chronic cardiac insufficiency (OR = 3.560 [1.955–6.482], p<0.001), previous diagnosis of arrhythmia (OR = 2.523 [1.436–4.434], p = 0.001), and previous history of ischemic heart failure (OR = 1.980 [1.012–3.873], p = 0.046). We also noted significant differences in patients taking oral anticoagulants (OR = 2.499 [1.415–4.415], p = 0.002), accented in patients treated with acenocoumarol (OR = 2.499 [1.704–5.610], p<0.001). On the other hand, we observed a decreasing incidence of daily aspirin intake (≤100mg/day) on the early mortality group, although no statistical significance was achieved (OR = 0.527 [0.254–1.093], p = 0.085).

We noted significant differences in patients with chronic obstructive pulmonary disease (OR = 2.096 [1.019–4.314], p = 0.044), and patients with a previous history of lung cancer (OR = 5.403 [1.065–27.413], p = 0.023), but no for other types of malignancy.

We also noted significant differences on ASA grade (OR = 3.273 [1.944–5.512], p<0.001) and Charlson comorbidity index (OR = 1.236[0.070–1.428], p = 0.004/High comorbidity (≥3) OR = 1.969 [1.132–3.422], p = 0.016).

## Postoperative clinical variables

**Postoperative surgical complications.**   Considering the arthroplasty patients—hemiarthroplasty and total hip arthroplasty, n = 407 –, we noted significant differences on early dislocation incidence for the 30-day mortality (OR = 6.557 [1.206–35.640], p = 0.029).

**Postoperative medical complications.**   We noted significant differences on many postoperative medical variables (OR = 4.050 [1.446–11.338], p = 0.008).

We noted significant differences on postoperative renal insufficiency (acute and flared-up, OR = 4.75 [2.446–9.205], p<0.001) and postoperative paralytic ileum (OR = 7.080 [1.77–28.174], p = 0.005).

We noted significant differences on cardiac complications (OR = 16.768 [9.052–31.062], p<0.001), including postsurgical arrhythmia (OR = 13.937 [6.263–31.017], p<0.001) and cardiac insufficiency (OR = 12.897 [6.525–25.488], p<0.001).

We also noted significant differences on respiratory complications (OR = 6.621 [3.769–11.631], p<0.001), including respiratory insufficiency (both acute and flared-up, OR = 7.002 [3.947–12.419], p<0.001) and upper respiratory infection (OR = 3.800 [1.499–9.635], p = 0.005).

We noted significant differences on sepsis (OR = 24.923 [4.075–152.436], p = 0.001), on postoperative nutritional supply requirement (OR = 3.576 [1.894–6.752], p = 0.021), and neurological complications (OR = 2.927 [1.629–5.258], p = 0.001), including postoperative delirium (OR = 2.565 [1.405–4.683], p = 0.002) and stroke (OR = 24.923 [4.075–152.436p<0.001).

We also noted statistical differences on metabolic ionic complications (OR = 3.287 [1.852–5.834], p<0.001), including sodium deregulation (OR = 2.929 [1.407–6.096], p = 0.004), hypernatremia (OR = 3.314 [1.092–10.056], p = 0.034) and hyponatremia (OR = 2.461 [0.997–6.076], p = 0.05), and potassium deregulation (OR = 5.238 [2.68–10.209], p<0.001), hyperkalaemia (OR = 10.378 [3.909–27.555], p<0.001) and hypokalaemia (OR = 2.934 [2.251–6.884], p = 0.013).

We also analysed in-hospital stay variables, noting a significant slight risk in the increasing length of stay (LOS) (OR per day = 1.091 [1.039–1.146], p = 0.001) and unveiling a protective factor for the walking ability at discharge (OR = 0.234 [0.127–0.432], p<0.001).

Fig 1 shows an odds ratio representation on early mortality of the mentioned significant variables.

At this stage within the study, we split the study population in half, by automatic randomization. It took 463 patients for multivariate risk assessment of selected preventable variables, and we validated the results for the 460 patients left. We could determinate a predictive model (S2 Table) that could explain 96.0% of non-early dying patients (sensitivity), and 60.7% of early dying patients (specificity), showing an AUC of 0.863 (Table 2, Fig 2).

The prediction equation for the developed model is as follows:

$$Prediction = \frac{1}{1 + e^{-SCORE}}$$

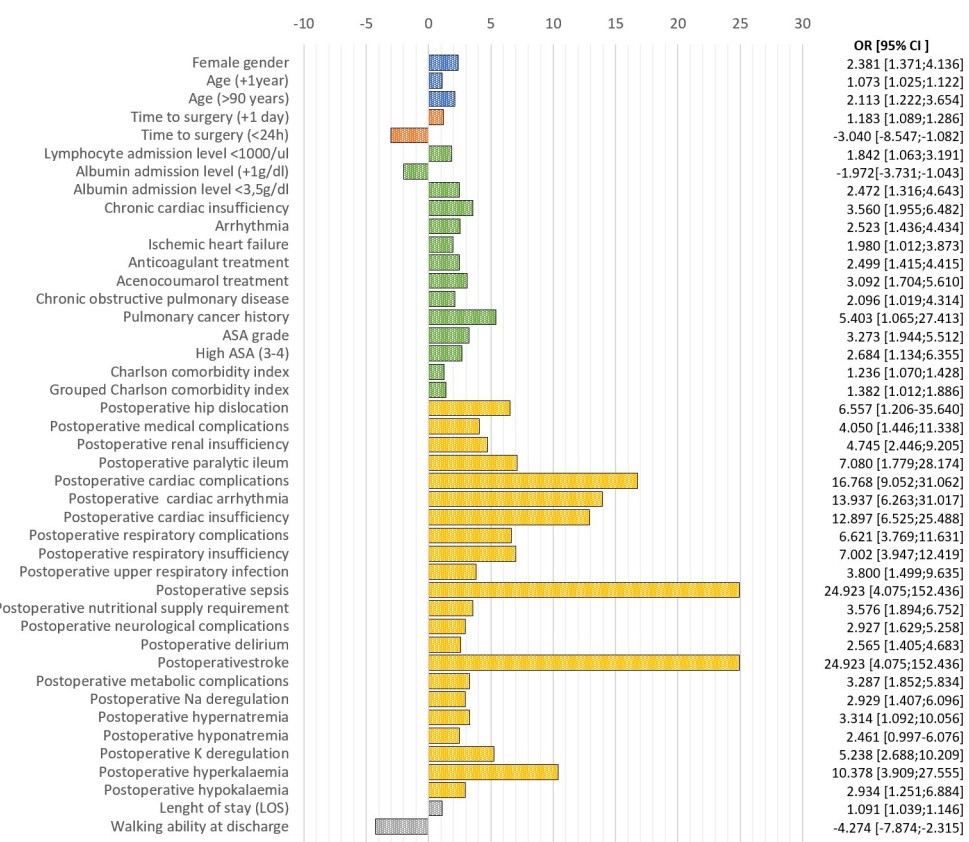

**Fig 1. Univariate risk representation of the studied variables showing statistical significance.** Blue: biodemographic variables; Red: fracture-related variables; Green: preoperative variables; Orange: postoperative variables; Grey: in-hospital stay derived variables.

**Table 2. Sensibility and specificity of the multivariate model on each stage of development.**

|  | Study population 1 n = 463 | Study population 2 n = 460 | Complete study population |
|---|---|---|---|
| Sensitivity | 98.9% | 95.2% | 96.0% |
| Specificity | 52.0% | 36.4% | 60.7% |

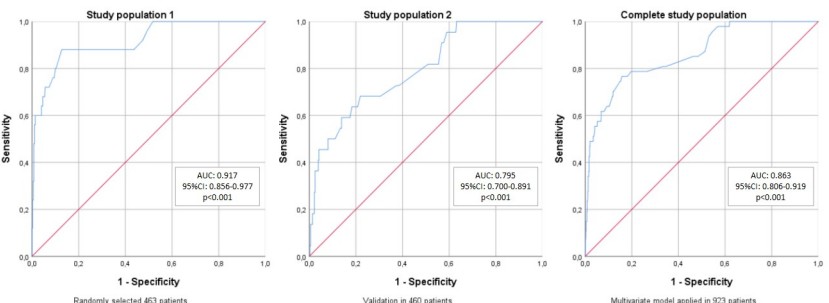

**Fig 2. ROC curves for the developed predictive model on early mortality after hip fracture surgery, on each stage of development.** AUC: Area under the curve; CI: Confidence interval.

**Table 3. Score calculation for the 30-day mortality prediction after hip fracture surgery in patients over 65.**

| Timeline | Variable | Record | SCORE |
|---|---|---|---|
| | Hip fracture surgical management | Yes | -3.931 |
| Upon hospital admission | Serum albumin (g/dl) | <3.5 | +1.479 |
| | | ≥3.5 | 0 |
| Early postoperative (before hospital discharge) | Nutritional supply requirement | Yes | +1.629 |
| | | No | 0 |
| | Acute confusional syndrome | Yes | +0.893 |
| | | No | 0 |
| | Hyperkalaemia | Yes | +1.757 |
| | | No | 0 |
| | Renal Insufficiency (acute or flared-up) | Yes | -0.906 |
| | | No | 0 |
| | Respiratory Insufficiency (acute or flared-up) | Yes | +1.429 |
| | | No | 0 |
| | Cardiac Insufficiency (acute or flared-up) | Yes | +2.568 |
| | | No | 0 |
| | Sepsis | Yes | +2.710 |
| | | No | 0 |
| | Stroke | Yes | +23.228 |
| | | No | 0 |
| | Walking ability | Yes | -1.597 |
| | | No | 0 |
| SCORE | Sum | | |

Table 3 enables the score calculation for the prediction equation application. The mathematical expression would be as follows:

$$SCORE = -3.931 + 1.479 \cdot LAlb_{AD} + 1.629 \cdot NU_{PO} + 0.893 \cdot ACS_{PO} + 1.757 \cdot AK^{+}{}_{PO} - 0.906 \\ \cdot RnI_{PO} + 1.429 \cdot RsI_{PO} + 2.568 \cdot CI_{PO} + 2.710 \cdot SP_{PO} + 23.228 \cdot ST_{PO} - 1.597 \cdot WA_{PO}$$

AD: at admission; PO: Postoperative. LAlb: Lower albumin (<3.5g/dl); WA: Walking ability; NU: nutritional supply requirement; ACS: Acute confusional syndrome; AK$^{+}$: Hyperkalaemia; RnI: Renal Insufficiency; RsI: Respiratory insufficiency; CI: Cardiac insufficiency; SP: Sepsis; ST: Stroke.

## Discussion

The current study analyses the influence of preoperative situation and comorbidities of older hip fracture patients on the 30-day mortality, but the main originality is that it also assesses potential early postoperative risk factors. We revealed that assessing certain postsurgical factors we could identify the frailest patients. While the postoperative risk factors assessed could be less modifiable, its identification itself represents a great advantage for the older hip fracture patients' management. The influence of post-operative factors on early mortality has a logical basis, albeit so far they have not been identified or quantified before. This insight would support the in-hospital patient's healthcare.

Mortality rates of hip fracture patients is a worldwide studied topic [3, 14–20]. The 30-day mortality rate after hip fracture surgery could be an indirect indicator of the in-hospital quality of care [4]. The mortality rates we report, were lower than the mean mortality rates from the Spanish National Hip Fracture Registry (SNHFR) [21], and lower than other international

studies, varying from 8.25% to 13.3% for the 30-day mortality [8, 11, 22]. These variations could reflect diverse factors, some would be inherent to each patient, like inner functional reserve, and others would be inherent to the healthcare process, as the time to surgery or the complications' management. Factors understanding could be important to concentrate efforts to avoid that excess early mortality.

## Biodemographic factors

Diverse studies have found a relationship between hip fracture mortality rate with age and gender [8, 23, 24], showing that the 30-day mortality rate after hip fracture is higher in men and oldest patients. We corroborate it, noting down the over 85 mean age of our population.

## Fracture-related factors

We found significantly increased time to surgery in patients who early died, what is consistent with many other studies backing up the idea that early surgery could avoid the early mortality [4, 22, 25–27]. However, it is difficult to consider the early surgery as a protective factor by itself, as this could not be achieved the more complex patients with poor previous medical conditions that may be playing as confounder factors. We consider maybe the risk factor previously assessed would be rather the late surgery than the early surgery. As our mean time to surgery never excess the five days, fewer patients underwent late surgery.

## Preoperative clinical factors

We noticed that the lower lymphocyte count and albumin admission levels were associated with the 30-day mortality. It could be also related to those patients who will require nutritional supply and reveal the importance of the good nutritional status of older patients [28, 29].

We also noted a negative effect of the anticoagulant treatment. Patients suffering from chronic cardiac insufficiency, previous diagnosis of arrhythmia, or previous history of ischemic heart failure use to take anticoagulant treatment. The anticoagulant treatment always carries a higher time to surgery and could be a reason why we noted an increased time to surgery on early dying patients. Rutenberg et al. [30] also showed a relationship between anticoagulant treatment and increased time to surgery for hip fracture, but they did not associate the anticoagulant treatment with the one-year mortality. Nonetheless, other previous studies also showed an increased time to surgery but also decreased survival [31–33].

Further, we recorded the Charlson comorbidity index, which showed a positive correlation with the 30-day mortality, as well as ASA grade. Both clinical scores assess high 30-day mortality in the worst clinical stated patients, what is consistent, and reveals the importance of the clinical status previous to hip fracture [24, 34].

## Postoperative clinical factors

Regarding the surgical-derived complications, we found the hip dislocation as a relevant risk factor for early mortality. Some authors already noted that most hip dislocations occurred within the first month after surgery [35–37]. Nonetheless, the incidence of this complication is varying from <1% to 3.5% [35, 37, 38]. Recently, Kishimoto et al. [39] pointed dislocation as a risk factor for long term survival after total hip arthroplasty, but only one case was a primary hip fracture. On the other hand, there are other works defeating no increase of mortality following a hip dislocation [35, 36]. Also, the prosthesis dislocation management may lead to bias for the results presented. Of the 407 patients who received hip arthroplasty in our study, only

seven patients suffered a hip dislocation. Taking account of the low incidence of this complication in our study, we would consider a greater population to validate this finding.

On the medical postoperative complications, sepsis was the analysed risk factor showing the highest OR. Although sepsis incidence was slight, it is concordant that patient's sepsis would carry a worse outcome. A similar upshot was revealed for the stroke [24]. The postoperative delirium is also a risk factor validated in this study, as previously pointed Mosk et al. [40] for later mortality.

We noted a higher incidence of sodium and potassium postoperative misbalance on the early mortality group, unexpected relevant risk factors not assessed before. It all raises the importance of metabolic monitoring for older patients undergoing hip fracture surgery. They could be also related to renal insufficiency, which also was assessed as a risk factor for early mortality, so those complications should be taken in high consideration, even with cardiac [41] and respiratory complications.

The ability to walk at discharge is a functional variable that seems to be a protective factor for the early mortality after hip fracture surgery, which agrees with its recent definition as a predictive factor for long-term survival [42].

Most studies already cited focused on pre-operative risk factors to assess the early mortality after hip fracture surgery, but we should consider the whole in-hospital process, including the immediate postoperative management, so it will significantly affect the patient's outcome. We developed a novel early mortality prediction model, in which we only included preventable variables. Preventable variables were defined as those variables that are not inherent to the patient, but physicians could modify and take into consideration in order to prevent or motivate. Our model has a 96.0% sensitivity and 60.7% specificity, and AUC of 0.863, which validates it for further consideration on larger populations. It would allow us to early detect the frailest patients who underwent hip fracture surgery so we could assess an earlier follow-up to prevent early mortality.

There are other recently developed models assessing early mortality after hip fracture. Karres et al. [11] evaluated six different predictive models, some of them extensively used (as NHFS or O-Possum), and none of them showed excellent discrimination, as their AUC always were less than 0.8. The most recently published early mortality predictive model for hip fracture is the Brabant Hip Fracture Score [12], but no postoperative approach was assessed.

In conclusion, we found significant preoperative risk factors for early mortality after hip fracture, but the postoperative risk factors revealed a higher impact on the patient's outcome at 30-days. In this sense, it is necessary to focus our efforts to decrease the postoperative complications rate on those patients in order to avoid the early mortality.

While the postoperative risk factors assessed could be less modifiable, its identification itself represents a great advantage for the older hip fracture patients' management. This insight would support the in-hospital patient's healthcare.

Besides our discussion, the study has some limitations as we studied single-center operated patients. Our upshot would be better considering a greater population.

## Supporting information

**S1 Table. Complete analysed descriptive data for all preoperative and postoperative factors analysed in the study population, and its comparison between the 30-day mortality group and survival group.**
(PDF)

**S2 Table. Definition of parameters in the equation for the predictive model assessing the 30-day mortality of hip fracture patients.**
(PDF)

## Author Contributions

**Conceptualization:** Juan F. Blanco.

**Data curation:** Carmen da Casa, Carmen Pablos-Hernández, Alfonso González-Ramírez, José Miguel Julián-Enríquez.

**Formal analysis:** Carmen da Casa, Agustín Díaz-Álvarez.

**Funding acquisition:** Juan F. Blanco.

**Investigation:** Carmen da Casa, Carmen Pablos-Hernández, Alfonso González-Ramírez, José Miguel Julián-Enríquez.

**Methodology:** Carmen da Casa, Agustín Díaz-Álvarez.

**Project administration:** Carmen da Casa.

**Resources:** Juan F. Blanco.

**Supervision:** Juan F. Blanco.

**Validation:** Carmen Pablos-Hernández, Alfonso González-Ramírez.

**Visualization:** Carmen da Casa.

**Writing – original draft:** Juan F. Blanco, Carmen da Casa.

**Writing – review & editing:** Juan F. Blanco, Carmen da Casa.

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
