## [Decision Letter · Decision Letter 0]

13 Jan 2021

PONE-D-20-36438

Early mortality after hip fracture surgery: influence of postoperative factors

PLOS ONE

Dear Dr. da Casa,

Thank you for submitting your manuscript to PLOS ONE. After careful consideration, we feel that it has merit but does not fully meet PLOS ONE’s publication criteria as it currently stands. Therefore, we invite you to submit a revised version of the manuscript that addresses the points raised during the review process.

We look forward to receiving your revised manuscript.

Kind regards,

Osama Farouk

Academic Editor

PLOS ONE

Journal Requirements:

Reviewers' comments:

Reviewer's Responses to Questions

**Comments to the Author**

1. Is the manuscript technically sound, and do the data support the conclusions?

Reviewer #1: Yes

Reviewer #2: Yes

2. Has the statistical analysis been performed appropriately and rigorously? 

Reviewer #1: Yes

Reviewer #2: Yes

3. Have the authors made all data underlying the findings in their manuscript fully available?

Reviewer #1: Yes

Reviewer #2: Yes

4. Is the manuscript presented in an intelligible fashion and written in standard English?

Reviewer #1: Yes

Reviewer #2: Yes

5. Review Comments to the Author

Reviewer #1: Dear authors, i appreciate the work done in this manuscript and i hope that my comments will help further improvement.

Title: i believe that it is better to mention that it is a 30 days mortality instead of just mentioning "early mortality"

Abstract: statistical analysis details better to be removed form the abstract, line 30: the word study is repeated, in the aim define what is considered as "early mortality"

Keywords: should include 30 days or at least early postoperative.

Introduction: well prepared and informative. the aim need to be more clear, define the early mortality (define the time period) as recommended earlier for the abstract.

-Methods: i would like the authors to clearly explain the aim of the telephonic follow up, did they only ask regarding mortality only, did they collect any data leading to mortality, if the patient is still alive would, how did the authors collected the data in concern?

-i understand that the aim was to detect the 30 days mortality, however, i was wondering why the authors chose to only evaluate the mortality in this short period, will patient surviving this period be able to live longer?

Results: the comparisons are not clear, it is confusing.

-Table 1 need to be more clear: what did the authors mean by rural residence, as this may differ between countries?, what did they mean by institution living (was it a nursing facility?)

-for the biodemographic variables, the authors mentioned that men had a higher early mortality, however in table 1 they reported that the female gender early mortality was 54.5% of the whole mortality incidence. The authors mentioned higher mortality in the 90 years age group, higher that which other group??

-for the fracture related variables: it is a common sense that patients who had a delayed surgery to have a comorbidity, as the medical issues in these fragile patients are the main reason for surgery delay, not just the fracture type. It is difficult to consider early surgery as a protective factor by itself , as this could not be achieved in all patients with medical comorbidities, higher ASA, bad lab variables are playing as a confounder for this variable.

-for the preoperative clinical values: it is better to divide these into categories (clinical, lab,...)

-postoperative clinical values: considering the type of surgery the patient had in this category is wrong, the other important point is the relation of the early prothesis dislocation to the early mortality, the authors should indicate if these patients who were subjected to early dislocations were treated surgically and were subjected to anathesia or not, as the dislocation by itself is not the risk, however, the way of management of this dislocation posses the risk on the patient.

-the postoperative category: better to be divided into subheadings for more fluency.

-Figures should be more clear

-Page 10, line 190: the authors mentioned "at this stage", what is the stage they refereed to at this point.

-page 10, line 198: the formula mentioned, score=: where is the results of this score, or the numbers the reader should get when applying the same formula. The formula mentioned is unclear and confusing, should be explained in a more simple and reproducible way.

Discussion: well presented.

Reviewer #2: The Manuscript:

Early mortality after hip fracture surgery: influence of postoperative factors

describes the results from a prospective cohort study on patients suffering from proximal Femur fractures. The authors identified several Risk factors and calculated a Risk scor System with a significant prediction rate. Principially, this is a very important work and the data is worthy for publication. In respect to several other epidemiological studies, probably some clarifications/idscussions wopuld be helpfull: the authors describe a Overall mortality of 6%, which might be normal in the analysed collective 65-100 years; hence, it would be good to calculate the "excess mortality" of the collective to identify the effect; moreover, the significant correlation between Prolongation of surgery above 24 or 48 Hours is well known as a Surrogate Parameter for co-morbidities, especially, anti-coagulative therapy. Hence, the conclusion to operate These patients as earloy as possible using potential dangerous pro-coagulative medication, such as PBSB, should be discussed critically according to quantitative Risk factor Adjustment. Besdie that, a good paper and congratultions

6. PLOS authors have the option to publish the peer review history of their article (what does this mean?). If published, this will include your full peer review and any attached files.

Reviewer #1: No

Reviewer #2: No

---

## [Author Response · Author response to Decision Letter 0]

15 Jan 2021

30-day mortality after hip fracture surgery: influence of postoperative factors. 

Response to reviwers

Reviewer #1: Dear authors, i appreciate the work done in this manuscript and i hope that my comments will help further improvement.

>Thank you very much. We hope that the arrangements you suggested and we made (further detailed) could improve the manuscript.

Title: i believe that it is better to mention that it is a 30 days mortality instead of just mentioning "early mortality"

>Actually we already had a talk about that topic and we agree with you and have changed the term also in the title.

Abstract: statistical analysis details better to be removed form the abstract, line 30: the word study is repeated, in the aim define what is considered as "early mortality"

>Thank you. We have corrected the erratum in text and specified “30-day mortality”.

Keywords: should include 30 days or at least early postoperative.

>Thank you, we have now included both terms as key words.

Introduction: well prepared and informative. the aim need to be more clear, define the early mortality (define the time period) as recommended earlier for the abstract.

>Thank you for your comment. We have enlarged line 68 “The early mortality rate, defined as 30-day mortality rate after hospital discharge, has been considered as an indirect indicator of the quality of care”, in order to be more informative and clear. We have also changed the term “early mortality” to “30-day mortality” in the purpose description.

-Methods: i would like the authors to clearly explain the aim of the telephonic follow up, did they only ask regarding mortality only, did they collect any data leading to mortality, if the patient is still alive would, how did the authors collected the data in concern?

>Indeed, at the 30-day follow-up some patients were invited to hospital visit and underwent a whole physical medical evaluation, but oldest or frailest patients were contacted by phone. At the telephonic follow-up we asked for the patient status, not only regarding mortality, but we only recorded mortality status, as it was the actual aim of the study data collection. We have no records on cause of mortality and, as we only designed a prospective study with 30-day follow-up after hospital discharge, we have no further record for longer mortality. 

We noted the day for the follow-up contact (at least 30-days after hospital discharge) and the dead/alive status of each patient. If the patient was already dead, we also noted the date for death, but no further records on leading cause for mortality were noted down.

See lines 96-97 “We performed on-site or telephonic 30-day follow up, noting down records for mortality.”

-i understand that the aim was to detect the 30 days mortality, however, i was wondering why the authors chose to only evaluate the mortality in this short period, will patient surviving this period be able to live longer?

>Thank you for the notation. We have already pointed it out on our previous response. Of course, patients could life much longer, indeed we have already published other studies designed with longer follow up [1]. For the present study, we aimed to note down the risk factors for the early mortality, what could be considered as indirect indicator of the healthcare quality during the hospital stay. The end-point of our work would be to allocate hospital resources according to patients’ requirements. Longer mortality could also be related to the patient inherent status and we should bear in mind that the mean age of the studied population is already over 85.

1. da Casa C, Pablos-Hernández C, González-Ramírez A, et al (2019) Geriatric scores can predict long-term survival rate after hip fracture surgery. BMC Geriatr 19:205. https://doi.org/10.1186/s12877-019-1223-y

Results: the comparisons are not clear, it is confusing.

>Thank you for the comment. We apologize for it. We have tried our best to improve results descriptions according to your recommendations. We have also changed Table 1 lines presentation (according to PLos One guidelines) to ease the statistical comparisons understanding.

-Table 1 need to be more clear: what did the authors mean by rural residence, as this may differ between countries?, what did they mean by institution living (was it a nursing facility?)

>Thank you again for your comment. We have specified both terms in Table 1 in order to be more clear and precise. Rural residence was considered for patients living in small municipalities composed by less than 12,500 inhabitants. They use to have a more active daily life that we though could be related to patient outcome. As you pointed, we meant patients living at a nursing facility when noting down “institution-living”; it was just a language misunderstanding since in Spain nursing homes are not the only ones in attending this type of institution-living patients (but also religious facilities in example).

-for the biodemographic variables, the authors mentioned that men had a higher early mortality, however in table 1 they reported that the female gender early mortality was 54.5% of the whole mortality incidence. The authors mentioned higher mortality in the 90 years age group, higher that which other group??

>Thank you for your comment. As you pointed, women represented over 50% of early deaths recorded, but we should bear in mind that women already represented over 70% of the whole study population. From another point of view, men were around 30% of the whole population, and around 50% of the early dying population. It all showed that a man being surgically treated for a hip fracture has increased probability for early dying than a woman. Results in Table 1 were presented for female gender as it is the bigger group of patients. In lines 128-130 we have tried to explain it “We noted a significantly higher rate of men in the early-mortality group […] than the original population gender distribution”.

We noted a “higher rate of patients over age 90 in the early mortality group”; higher than patients under 90 years’ age group. We apologize if it was not enough clearly exposed.

-for the fracture related variables: it is a common sense that patients who had a delayed surgery to have a comorbidity, as the medical issues in these fragile patients are the main reason for surgery delay, not just the fracture type. It is difficult to consider early surgery as a protective factor by itself, as this could not be achieved in all patients with medical comorbidities, higher ASA, bad lab variables are playing as a confounder for this variable.

>Thank you again for your comment. We agree with you, as we already pointed in our discussion: “We consider maybe the risk factor assessed would be rather the late surgery than the early surgery.”

We have no further included the early surgery on the statistical model developed, due to, as you pointed, it could not be modifiable in the more complex patients. We hope you have no objection that we have incorporated your comment with slight modifications to the revised version of our manuscript.

-for the preoperative clinical values: it is better to divide these into categories (clinical, lab,...)

>Thank you for the notation. We have arranged it.

-postoperative clinical values: considering the type of surgery the patient had in this category is wrong, the other important point is the relation of the early prothesis dislocation to the early mortality, the authors should indicate if these patients who were subjected to early dislocations were treated surgically and were subjected to anathesia or not, as the dislocation by itself is not the risk, however, the way of management of this dislocation posses the risk on the patient.

>Thank you for your comment. We considered the type of surgery as a “fracture-related” factor (see lines 135-136), but the early prosthesis dislocation was considered as a postoperative clinical factor, only noted for arthroplasty-treated patients (osteosynthesis-based patients could not undergo prosthesis dislocation and were excluded for statistical analysis). We acknowledge your contribution regarding the dislocation management. Unfortunately, we have no records on it, and the study stood only for the dislocation event itself. Again, we should bear in mind that the incidence of prosthesis dislocation is low, and, although we studied a large population, it is still small to be able to split the prosthesis dislocation management and study its consequences. We have annotated your contribution on the revised manuscript (line 298). 

-the postoperative category: better to be divided into subheadings for more fluency.

>Thank you for the notation. We have tried to arranged it by adding two subheadings and splitting paragraphs in order to expose it easier for lectors. 

-Figures should be more clear

>Thank you for the comment. We have improved the images quality and we hope they would be now clearer for publication.

-Page 10, line 190: the authors mentioned "at this stage", what is the stage they refereed to at this point.

>Thank for the comment. Probably we just omit what we meant. It was just a conjunction for expressing the study stage. We firstly made the univariate analysis and then developed the predictive model.

-page 10, line 198: the formula mentioned, score=: where is the results of this score, or the numbers the reader should get when applying the same formula. The formula mentioned is unclear and confusing, should be explained in a more simple and reproducible way.

>Thank you for the comment. We have tried our best to clarify the prediction calculation. We have translated the complex mathematical formulae to a simpler and useful table (see Table 3) in order to obtain the score. We acknowledge your contribution and we believe that with the table format would be easier to apply the prediction model here presented. 

Discussion: well presented.

>Thank you very much. We appreciate your time and effort and we believe your notations have help us to improve our manuscript.

Reviewer #2: 

The Manuscript: Early mortality after hip fracture surgery: influence of postoperative factors

describes the results from a prospective cohort study on patients suffering from proximal Femur fractures. The authors identified several Risk factors and calculated a Risk scor System with a significant prediction rate. Principially, this is a very important work and the data is worthy for publication. 

>Thank you very much for your appreciation.

In respect to several other epidemiological studies, probably some clarifications/idscussions wopuld be helpfull: the authors describe a Overall mortality of 6%, which might be normal in the analysed collective 65-100 years; hence, it would be good to calculate the "excess mortality" of the collective to identify the effect;

>Thank you for your comment. We really appreciate your suggestion, but we certainly don't have the data nor the necessary tools to be able to carry out the discussion you are suggesting. It would be probably a great complementary study that we will further try to perform. Unfortunately, at the moment, we are unable to present the requested calculation.

moreover, the significant correlation between Prolongation of surgery above 24 or 48 Hours is well known as a Surrogate Parameter for co-morbidities, especially, anti-coagulative therapy. Hence, the conclusion to operate These patients as earloy as possible using potential dangerous pro-coagulative medication, such as PBSB, should be discussed critically according to quantitative Risk factor Adjustment. 

>Thank you for your notation. We agree with you and we have tried to include this statement on the discussion (Lines 166-168). We have no records on the use of pro-coagulative medication in this study population, but we will bear it in mind for further studies. Nevertheless, we did not to include the early surgery on the prediction model developed as it could not be modifiable in the more complex patients.

Besdie that, a good paper and congratulations

>Thank you very much again for your appreciation. We hope that all other arrangements we made could improve our manuscript.

---

## [Decision Letter · Decision Letter 1]

29 Jan 2021

30-day mortality after hip fracture surgery: influence of postoperative factors

PONE-D-20-36438R1

Dear Dr. da Casa,

We’re pleased to inform you that your manuscript has been judged scientifically suitable for publication and will be formally accepted for publication once it meets all outstanding technical requirements.

Kind regards,

Osama Farouk

Academic Editor

PLOS ONE

Additional Editor Comments (optional):

Reviewers' comments:

Reviewer's Responses to Questions

**Comments to the Author**

1. If the authors have adequately addressed your comments raised in a previous round of review and you feel that this manuscript is now acceptable for publication, you may indicate that here to bypass the “Comments to the Author” section, enter your conflict of interest statement in the “Confidential to Editor” section, and submit your "Accept" recommendation.

Reviewer #1: All comments have been addressed

Reviewer #2: All comments have been addressed

2. Is the manuscript technically sound, and do the data support the conclusions?

Reviewer #1: Yes

Reviewer #2: Yes

3. Has the statistical analysis been performed appropriately and rigorously? 

Reviewer #1: Yes

Reviewer #2: Yes

4. Have the authors made all data underlying the findings in their manuscript fully available?

Reviewer #1: Yes

Reviewer #2: Yes

5. Is the manuscript presented in an intelligible fashion and written in standard English?

Reviewer #1: Yes

Reviewer #2: Yes

6. Review Comments to the Author

Reviewer #1: Dear authors, i was very delighted to review your revised version of the manuscript, thanks for responding to the recommendations and suggestions. Wish you all the best.

Reviewer #2: Although the data of excess mortality could not be provided, I think this is an improtant piece of work for the scientific community and should be published

7. PLOS authors have the option to publish the peer review history of their article (what does this mean?). If published, this will include your full peer review and any attached files.

Reviewer #1: **Yes: **Ahmed Adel Khalifa, MD, FRCS, MSc

Reviewer #2: No

---

## [Editor Report · Acceptance letter]

2 Feb 2021

PONE-D-20-36438R1 

30-day mortality after hip fracture surgery: influence of postoperative factors. 

Dear Dr. da Casa:

I'm pleased to inform you that your manuscript has been deemed suitable for publication in PLOS ONE. Congratulations! Your manuscript is now with our production department. 

Kind regards, 

on behalf of

Dr. Osama Farouk 

Academic Editor

PLOS ONE